Validation of the Emotiv EPOC EEG system for research quality auditory event-related potentials in children

Badcock Nicholas A. 1 2 5 nicholas.badcock@mq.edu.au
Preece Kathryn A. 1 2 5
de Wit Bianca 1 2
Glenn Katharine 3
Fieder Nora 1 2
Thie Johnson 4
McArthur Genevieve 1 2
1 ARC Centre of Excellence in Cognition and its Disorders, Macquarie University , North Ryde, NSW , Australia
2 Department of Cognitive Science, Macquarie University , North Ryde, NSW , Australia
3 MultiLit , Macquarie Park, NSW , Australia
4 School of Electrical and Information Engineering, University of Sydney , Sydney, NSW , Australia
Abdullah Jafri
5 Joint first authors for this work.

Electronic publication date: 2015 Apr 21
Publication date: 2015
Volume: 3
Electronic Location ID: e907
Received 2014 Nov 6; Accepted 2015 Apr 1
Copyright: © 2015 Badcock et al.
Copyright year: 2015
Copyright holder: Badcock et al.
License: This is an open access article distributed under the terms of the Creative Commons Attribution License, which permits unrestricted use, distribution, reproduction and adaptation in any medium and for any purpose provided that it is properly attributed. For attribution, the original author(s), title, publication source (PeerJ) and either DOI or URL of the article must be cited.
License URL: https://creativecommons.org/licenses/by/4.0/

Keywords: EEG, ERP, Emotiv EPOC, Validation, Mismatchnegativity, MMN, Intraclass correlation, Methods, Auditory odd-ball, Children

Funding: ARC Centre of Excellence CE110001021 This research was supported by an ARC Centre of Excellence Grant [CE110001021] and an NHMRC equipment grant. The funders had no role in study design, data collection and analysis, decision to publish, or preparation of the manuscript.

==============================
Background. Previous work has demonstrated that a commercial gaming electroencephalography (EEG) system, Emotiv EPOC, can be adjusted to provide valid auditory event-related potentials (ERPs) in adults that are comparable to ERPs recorded by a research-grade EEG system, Neuroscan. The aim of the current study was to determine if the same was true for children.

Method. An adapted Emotiv EPOC system and Neuroscan system were used to make simultaneous EEG recordings in nineteen 6- to 12-year-old children under “passive” and “active” listening conditions. In the passive condition, children were instructed to watch a silent DVD and ignore 566 standard (1,000 Hz) and 100 deviant (1,200 Hz) tones. In the active condition, they listened to the same stimuli, and were asked to count the number of ‘high’ (i.e., deviant) tones.

Results. Intraclass correlations (ICCs) indicated that the ERP morphology recorded with the two systems was very similar for the P1, N1, P2, N2, and P3 ERP peaks (r = .82 to .95) in both passive and active conditions, and less so, though still strong, for mismatch negativity ERP component (MMN; r = .67 to .74). There were few differences between peak amplitude and latency estimates for the two systems.

Conclusions. An adapted EPOC EEG system can be used to index children’s late auditory ERP peaks (i.e., P1, N1, P2, N2, P3) and their MMN ERP component.

Introduction

An auditory event-related potential (ERP) is the average pattern of electrical activity generated by neurons in response to a particular auditory event. Auditory ERPs can be measured without a listener’s overt attention. Such “passive” auditory ERPs are a useful means of investigating the role of auditory processing in people who find it difficult to pay attention to stimuli, to make decisions about stimuli, or plan overt responses to stimuli. Thus, passive auditory ERPs have proved useful for investigating auditory processing in attention deficit hyperactivity disorder (ADHD; Taylor et al., 1997), schizophrenia (Todd, Michie & Jablensky, 2003); autism (McPartland et al., 2004); developmental dyslexia (McArthur, Atkinson & Ellis, 2009); and specific language impairment (Whitehouse, Barry & Bishop, 2008).

A limitation of passive auditory ERPs is that they are typically measured using research-grade equipment housed in a laboratory. Such settings can be intimidating for many people, particularly children and adults with cognitive disorders. Fortunately, recent research has shown that a commercial “gaming” electroencephalography (EEG) system, called “EPOC” by Emotiv (www.emotiv.com), can be adapted to produce valid ERPs. Badcock et al. (2013) examined auditory ERPs in “passive” (standard and deviant tones are ignored) and “active” (deviant tones are counted) listening conditions in adults, using an adapted EPOC system and a research-grade Neuroscan system. They found high reliability for the “late auditory ERP” peaks (i.e., P1, N1, P2, N2, and P3) but not for the “mismatch negativity” component (MMN; see Näätänen et al., 2004). The EPOC system has also been successfully used to measure the auditory P3 response (Debener et al., 2012; De Vos, Gandras & Debener, 2014) and the visual P3 response (Duvinage et al., 2013; De Vos et al., 2014). Considered together, the outcomes of these seminal studies suggest that the EPOC system can be adapted to record valid auditory P1, N1, P2, N2, and P3 ERP peaks in adults.

To our knowledge, no study has yet tested if an adapted EPOC system can produce valid auditory ERPs in children. This cannot be inferred from previous validation studies done with adults because children have (1) different ERPs to adults due to cortical and cognitive immaturity (Ponton et al., 2002; Coch, Sanders & Neville, 2005; Mahajan & McArthur, 2012; Mahajan & McArthur, 2013); (2) “noisier” ERPs than adults (Coch & Gullick, 2012); and (3) more difficultly keeping still during long test sessions than adults, so their EEG (and ERP) responses may be contaminated to a greater degree by electrical noise generated by movement. The aim of the current study was to test the validity of children’s passive and active auditory ERPs measured via an adapted EPOC system. In line with an analogous adult study (i.e., Badcock et al., 2013) we predicted that the adapted EPOC system would produce valid ERPs for the highly reliable late auditory ERP peaks (P1, N1, P2, P2, P3) but invalid ERPs for the less reliable MMN component.

Materials and Methods

The Macquarie University Human Research Ethics Committee approved the methods used in this study (approval number: 5201200658).

Participants

Participants were twenty-one children (11 females, 10 males) aged between 6 and 12 years (M = 9.23, SD = 1.80). Parents or guardians of the children provided written informed consent for their child’s participation (see Supplemental Information for information-consent form), and children were reimbursed $15 for their time. Participants were required to have normal hearing and vision, and no history of epilepsy. One child was excluded from the study due to a reported hearing loss; another child was excluded because the EPOC event-markers failed to record; and a third was excluded due to a condition coding error. Therefore, the final sample included 18 children. Based on our previous research demonstrating a large effect size (0.7, Badcock et al., 2013), G*Power (Faul et al., 2007; Faul et al., 2009) estimated that a sample size of 11 would provide adequate power for this experiment (0.8).

Stimuli

Presentation (Version 16; Neurobehavioural Systems) was used to deliver tones in passive and active conditions (see below) at a volume that was comfortable for each participant (note: the volume remained fixed across conditions). Each condition consisted of 566, 175-ms, 1000-Hz standard tones (10-ms rise and fall time; 85% of trials) and 100, 175-ms, 1200-Hz deviant tones (10-ms rise and fall time; 15% of trials). Deviant tones were presented after 3 to 35 (randomly allocated) standard tones. The stimulus onset asynchrony was jittered between 900 and 1100 ms to minimize EEG activity related to anticipatory processes. Tones were presented binaurally via Phillips SHS4700/37 ear clip headphones (Phillips, Amsterdam, Netherlands) fixed to the EPOC headset.

In the passive condition, participants were instructed to watch a silent movie and ignore the tones presented through the headphones. In the active condition, participants were instructed to count the “high” tones whilst watching the silent movie. Participants were asked, and reminded where necessary, to stay as still as possible. Each condition lasted approximately 13 min, separated by a short-break.

Neuroscan system

The research-grade EEG system (Neuroscan Version 4.3; widely accepted for psychophysiological research) used an EEG electrode cap (EasyCap, Herrsching, Germany) fitted with 14 Ag-AgCl electrodes located at F3, F7, FC4, FT7, T7, P7, P8, T8, FT8, FC4, F8, F4, M1 (online reference), and M2. Electrodes placed above and below the left eye measured vertical eye movements (“VEOG”), and electrodes placed on the outer side of each eye measured horizontal eye movements (“HEOG”). Please note that the M2 (right mastoid), VEOG, and HEOG electrodes were set up as per standard procedures even though these electrodes were not used in the analysis as EPOC does not provide equivalent measurements. The ground electrode was positioned between FPz and Fz.

Neuroscan was recorded at 1000 Hz. Triggers were inserted into the EEG to indicate the onset of each stimulus. These triggers were generated by Presentation, and were inserted into the EEG via a parallel port.

Adapted EPOC system

The EPOC system used a wireless headset with flexible plastic arms that held gold-plated sensors against the head at 16 sites that aligned with the research EEG headset: AF3, F7, F3, FC5, T7, P7, O1, O2, P8, T8, FC6, F4, F8, AF4, M1 and M2. M1 acted as a ground reference point for measuring the voltage of the other sensors. M2 acted as a feed-forward reference point for reducing electrical interference from external sources. The remaining signals were high-pass filtered with a 0.16 Hz cut-off, pre-amplified and low-pass filtered at an 83 Hz cut-off. The analogue signals were then digitised at 2048 Hz, filtered using a 5th-order sinc notch filter (50–60 Hz), and low-pass filtered before being down-sampled to 128 Hz (specifications taken from the gaming EEG system web forum). The effective bandwidth was 0.16–43 Hz. EPOC was recorded at 128 Hz.

The EPOC system was adapted to accurately time-lock the EEG signal to the onset of each stimulus by marking the EEG signal with an electrical pulse triggered by a wireless transmission system (Thie, 2013). The system consisted of transmitter and receiver units that were linked using infrared (IR) light. The transmitter unit was attached to the audio output of the stimulus presentation computer. The receiver unit was mounted in close proximity to the participant (i.e., taped to their shoulder or resting on a table) with its output wires attached to two of the EEG electrodes (O1 and O2). These electrodes were attached directly to the Driven Right Leg (DRL) through wires and 4700-ohm resistors that mimicked a perfect connection with the scalp. The transmitter unit was made up of a microcontroller board (Arduino Uno) and an interface board. This board amplified the audio stimuli and fed it to the Arduino’s analogue input. The receiver waited for a number from the transmitter to trigger a 100-ms-wide pulse. There was a 19-ms delay (accounted for the trigger processing) between the onset of the stimulus and the onset of the marker pulse due to the buffering of the audio signal in order to determine its frequency and the transmission of the 8-bit number.

Procedure

Neuroscan was setup first and adjusted until sensor impedance was below 5 kOhms. The EPOC headset was fitted over the EasyCap (for a detailed description, see Badcock et al., 2013). This allowed for simultaneous measurements of EEG by the Neuroscan and EPOC systems (see Fig. 1; for electrode locations see Fig. 2). EPOC electrode connectivity was tested using the TestBench software. Sensors were adjusted until connectivity reached the “green” level, which represented impendences less than 220 kOhms (measured using a resistor between an electrode and the DRL, M2 in the current setup). The total setup time was approximately 55 min.

Figure 1 Schematic diagram of simultaneous Neuroscan (in grey) and EPOC (in black) setup, including infrared transmission for EPOC event markers.

Figure 2 Schematic diagram depicting the placement of EEG electrodes for Neuroscan (blue targets) and EPOC (orange crosses) systems.

Offline EEG processing

Both Neuroscan and EPOC EEG recordings were processed in the same way using EEGLAB version 11.0.4.3b (Delorme & Makeig, 2004). Large artefacts in each EEG were first excluded by eye. The Neuroscan EEG data were then downsampled to 128 Hz in order to match the sampling rate of the EPOC system. The EEG data were then bandpass filtered from 0.1 to 30 Hz, separated into epochs that started −102 ms before the onset of each tone and ended 500 ms after the onset of each tone, and baseline corrected between −102 and 0 ms. Any epochs with an amplitude in excess of ±150 µV were excluded.

Ocular artefact removal was attempted using Independent Components Analysis in EEGLAB (note: channels capturing the eye-movements for Neuroscan were not included in this process to maintain equivalent processing between the systems). This process did not identify eye-blink related components in any of the datasets. Therefore, eye-blinks were either not consistent or strong enough to meaningfully affect the data.

In order to best compare the two systems, only those epochs accepted for both EEG systems (i.e., shared epochs) were included in the analysis (we thank Phillip Ruhnau for this suggestion). For each child, the shared epochs were averaged together to produce late auditory ERP waveforms that comprised P1, N1, P2, and N2 peaks for the standard and deviant tones, in the passive and active conditions. Shared epochs to standard and deviant tones in the passive condition were averaged separately and then subtracted (i.e., the ERP to standard tones was subtracted from the ERP to deviant tones) to produce a mismatch negativity (MMN) waveform.

Analysis

In line with the previous EPOC validation study done with adults, the analysis focused on data from frontal sites in the left and right hemispheres: F3 and F4 for Neuroscan, and AF3 and AF4 for EPOC.

The ERPs produced by the two systems were compared in three ways: (1) total number of accepted epochs were used to compare the quality of the Neuroscan and EPOC EEG data, (2) intraclass correlations (ICCs, for details see McArthur & Bishop, 2004; Bishop et al., 2007) were used to index the similarity of Neuroscan and EPOC waveforms (between −102 to 500 ms), and (3) peak amplitude and latency measures were used to compare the size and timing of each ERP peak or component. The number of epochs and peak comparison data sets were tested for normality (Shapiro–Wilk) and equal variance (F test). Single- and paired-sample t-tests and Wilcoxon-signed ranks were used to evaluate the statistical reliability between EEGs systems comparisons, and Cohen’s d was used to evaluate the magnitude of the effects. We used a criteria of p < .05 unless otherwise specified.

Regarding the size and timing of ERP peaks and components, peak amplitude and latency measures were initially calculated using an automated procedure that identified the point of maximum amplitude (positive or negative) within appropriate time intervals or “search windows.” These intervals were determined by visual inspection of the relevant grand mean ERP waveforms (as described by Hoormann et al., 1998), and were as follows: 50 to 140 ms (P1); 70 to 140 (N1); 140 to 200 ms (P2); 260 to 400 ms (N2); 260 to 400 ms (P3); 140 to 260 (MMN). We then checked the validity of each peak measure for each child by visually inspecting individual waveforms. This revealed that the N1 and P2 peaks were missing in 11 to 14 (60 to 77%) children across all conditions, which is characteristic of children’s auditory ERPs (Ponton et al., 2000; Mahajan & McArthur, 2012; Mahajan & McArthur, 2013). Invalid data points for missing peaks were deleted from the dataset. A further 15% of the measures produced by the automated peak detection were invalid, identifying an end-point of the range greater in magnitude that the true peak. Invalid end-point measures were corrected manually to ensure all peak amplitude and latency measures for all children were valid.

Results

Number of accepted epochs

The distributions for the number of accepted epochs were negatively skewed, thus, Wilcoxon Signed Rank Tests were used to compare the data recorded by the two systems. The median number of accepted epochs, inter-quartile range, and Wilcoxon signed ranks statistics are presented in Table 1. There were statistically fewer acceptable epochs for EPOC than Neuroscan in all conditions. Nevertheless, the number of accepted epochs for both the EPOC and Neuroscan systems was more than adequate for waveform generation for all participants.

Table 1 Median number of accepted and shared epochs for Neuroscan and EPOC by condition and tone type.

Median (inter-quartile range) number of accepted epochs for the Neuroscan and EPOC systems and total number of epochs shared (i.e. accepted for both systems) between systems in each condition (passive and active) for each tone type (standard, deviant, and total). Wilcoxon Signed Rank Tests (Z) were used to test the difference between systems.

		EEG system			
Condition	Tone	Neuroscan	EPOC	Z	Shared epochs	
Passive	Standard	558 (18)	528 (36)	3.70*	523 (38)	
	Deviant	98 (4)	94 (8)	3.39*	94 (9)	
	Total	656 (23)	622 (37)		614 (39)	
Active	Standard	558 (18)	518 (36)	3.66*	517 (34)	
	Deviant	98 (3)	92 (4)	3.60*	92 (4)	
	Total	656 (22)	610 (38)		608 (39)	
Notes.

* p < .05

ICCs

P1, N1, P2, and N2

The mean of the group ERP waveforms produced by the Neuroscan and EPOC systems to the standard and deviant tones in the passive and active conditions are displayed in Fig. 3 (see Figs. S1 and S2 for the auditory ERPs of individual children). The ICCs between late auditory ERP waveforms generated by the two systems to standard and deviant tones in the passive and active conditions are presented in Table 2. The range of ICCs for the standard tones was 0.91 to 0.95 and for the deviant tones was 0.82 to 0.86. All of these distributions were negatively skewed; therefore, statistical differences to zero were assessed using single-sample Wilcoxon signed ranks, all of which were significant: all Z = 4.48, p < .001. These results indicate a strong correspondence between the measurements made with the two systems.

Figure 3 Event-related potential (ERP) waveforms for Neuroscan and EPOC by tone, hemisphere, and condition.

All graphs display the group average ERP waveforms for the passive (ignore tones) and active (count deviant tones) listening conditions. Data collected with Neuroscan and EPOC are presented in the left and right columns respectively, ERPs to the standard (low) tones are presented in panels A, B, E, & F, and ERPs to the deviant (high) tones are presented in panels C, D, G, & H. The upper four panels depicted ERPs from the left-hemisphere (Neuroscan = F3: A & C; EPOC = AF3: B & D) and the lower four panels depicted ERPs from the right-hemisphere (Neuroscan = F4: E & G; EPOC = AF4: F & H).

Table 2 Neuroscan versus EPOC ERP and MMN waveform intraclass correlations.

Mean intraclass correlations (ICC) (with 95% confidence intervals) between Neuroscan and EPOC late auditory P1, N1, P2, N2, and P3 ERPs and the MMN component at F3/AF3 and F4/AF4 to standard and deviant tones in both passive and active conditions. Single-sample Wilcoxon signed rank test p-values are represented.

		Electrode	
Condition	Tone	F3/AF3	F4/AF4	
Passive	Standard	0.95 [0.93, 0.97]	0.93 [0.91, 0.95]	
	Deviant	0.86 [0.81, 0.91]	0.82 [0.74, 0.90]	
	MMN	0.74 [0.65, 0.83]	0.67 [0.56, 0.78]	
Active	Standard	0.94 [0.92, 0.96]	0.91 [0.88, 0.94]	
	Deviant	0.85 [0.79, 0.91]	0.83 [0.77, 0.89]	
Notes.

All p < .001.

P3

The mean of the group late auditory ERP waveforms produced by the Neuroscan and EPOC systems to the deviant tones in the active condition are displayed in Fig. 3 (see Fig. S2 for the auditory ERPs of individual children). The corresponding ICC values are shown in Table 2. The ICCs for F3/AF3 and F4/AF4 were 0.85 and 0.83 respectively, and the negatively skewed distributions were significantly different to zero: both single-sample Wilcoxon signed ranks, Z = 4.48, p < .001. These results indicate a strong correspondence between the measurements made with the two systems.

MMN

The mean of the group MMN ERP waveforms produced by the Neuroscan and EPOC systems are presented in Fig. 4 (see Fig. S3 for the MMN ERP waveforms of individual children). The ICCs between the MMN waveforms generated by the Neuroscan and EPOC systems are shown in Table 4. These ICCs were: 0.74 for F3/AF3, and 0.67 for F4/AF4. Both distributions were normally distributed and single-sample t-tests determined the ICCs were statistically different to zero; F3/AF4 , t(17) = 15.28, p < .001; F4/AF4, t(17) = 12.02, p < .001. These results indicate a moderate to strong correspondence for the MMN waveforms between the measurements made with the two systems.

Figure 4 Event-related potential (ERP) and Mismatch Negativity (MMN) waveforms for Neuroscan and EPOC by tone and hemisphere for the passive condition (ignore tones).

Data collected with the Neuroscan and EPOC are presented in the left and right columns respectively, ERPs to the standard (low) and deviant (high) tones are presented in panels A, B, E, & F, and the difference between these waveforms (i.e., the mismatch negativity responses) are presented in panels C, D, G, & H. The upper four panels depicted ERPs from the left-hemisphere (Neuroscan = F3: A & C; EPOC = AF3: B & D) and the lower four panels depicted ERPs from the right-hemisphere (Neuroscan = F4: E & G; EPOC = AF4: F & H).

Peak amplitude and latency

The descriptive statistics for P1, N1, P2, N2, P3 and MMN peak amplitude and latency measures produced by the Neuroscan and EPOC systems for standard and deviant tones in the passive and active conditions at F3/AF4 and F4/AF4 are reported in Tables 3–5. Peak comparisons between the two systems were conducted using paired-samples t-tests and Wilcoxon singed rank tests, depending upon the normality of the data as indicated in the tables. Due to multiple comparisons, statistical tests with p-values less than .01 will be highlighted (p < .05 and .001 are also indicated in the tables).

Table 3 Neuroscan versus EPOC ERP peak comparisons: passive listening.

Descriptive (n, M [lower, upper 95% confidence intervals]) and inferential (t or Z and Cohen’s d) statistics for peak (P1, N1, P2, N2) amplitude (µV) and latency (ms) measures at sties F3/AF3 and F4/AF4 for Neuroscan versus EPOC in the passive condition.

					EEG system			
Tone	ERP	Measure	Electrode	n	Neuroscan	EPOC	stat.	d	
Standard	P1	Amplitude	F3/AF3	18	3.44 [2.3, 4.6]	3.36 [2.1, 4.6]	−0.17a	0.03	
			F4/AF4	18	3.66 [2.7, 4.6]	3.39 [2.3, 4.5]	1.12	0.13	
		Latency	F3/AF3	18	98 [89, 107]	104 [96, 112]	−3.39a,***	0.31	
			F4/AF4	18	99 [90, 108]	104 [95, 113]	−2.74a,**	0.29	
	N1	Amplitude	F3/AF3	9	−0.83 [−2.0, 0.4]	−1.00 [−2.0, 0.0]	0.6	0.11	
			F4/AF4	9	−0.97 [−1.9, −0.0]	−1.10 [−2.1, −0.1]	0.52	0.09	
		Latency	F3/AF3	9	122 [114, 130]	131 [122, 140]	−3.77**	0.81	
			F4/AF4	9	119] [111, 127]	128 [122, 134]	−5.54***	0.89	
	P2	Amplitude	F3/AF3	9	1.34 [−0.7, 3.4]	1.11 [−0.6, 2.8]	0.64	0.09	
			F4/AF4	9	1.83 [−0.3, 4.0]	1.32 [−0.3, 2.9]	1.06	0.19	
		Latency	F3/AF3	9	152 [144, 160]	162 [154, 170]	−4.63**	0.93	
			F4/AF4	9	155 [146, 164]	169 [159, 179]	−4.06**	1.09	
	N2	Amplitude	F3/AF3	18	−9.32 [−10.9, −7.7]	−8.33 [−10.0, −6.6]	−3.59a,***	0.29	
			F4/AF4	18	−8.80 [−10.6, −7.0]	−8.65 [−10.2, −7.1]	−0.49	0.04	
		Latency	F3/AF3	18	268 [258, 278]	276 [266, 286]	−4.88***	0.41	
			F4/AF4	18	265 [256, 274]	278 [268, 288]	−4.03***	0.63	
Deviant	P1	Amplitude	F3/AF3	18	3.90 [2.4, 5.4]	3.68 [2.1, 5.2]	−0.55a	0.07	
			F4/AF4	18	3.88 [2.6, 5.2]	3.42 [1.9, 4.9]	−1.07a	0.16	
		Latency	F3/AF3	18	95 [87, 103]	103 [94, 112]	−3.59a,***	0.51	
			F4/AF4	18	97 [88, 106]	99 [89, 109]	−0.64	0.13	
	N1	Amplitude	F3/AF3	7	−2.01 [−3.1, −0.9]	−2.14 [−3.4, −0.9]	0.36	0.09	
			F4/AF4	7	−1.11 [−2.5, 0.3]	−2.51 [−3.9, −1.2]	−1.6a	0.86	
		Latency	F3/AF3	7	133 [117, 149]	131 [123, 139]	0.31	0.13	
			F4/AF4	7	125 [115, 135]	124 [104, 144]	0.1	0.05	
	P2	Amplitude	F3/AF3	7	1.75 [−0.4, 3.9]	0.81 [−0.8, 2.4]	0.95	0.41	
			F4/AF4	7	1.38 [−1.5, 4.3]	1.27 [−0.5, 3.1]	0.11	0.04	
		Latency	F3/AF3	7	160 [148, 172]	167 [155, 179]	−2.55a,*	0.52	
			F4/AF4	7	166 [152, 180]	163 [148, 178]	0.39	0.17	
	N2	Amplitude	F3/AF3	18	−10.30 [−12.3, −8.3]	−9.57 [−11.5, −7.6]	−1.69	0.18	
			F4/AF4	18	−10.10 [−11.8, −8.4]	−9.87 [−11.1, −8.6]	−0.49	0.07	
		Latency	F3/AF3	18	227 [215, 239]	237 [225, 249]	−2.71a,**	0.38	
			F4/AF4	18	227 [215, 239]	235 [225, 245]	−2.25a,*	0.33	
Notes.

a Wilcoxon Z.

* p < .05.

** p < .01.

*** p < .001.

Table 4 Neuroscan versus EPOC EEG system ERP peak comparisons: active listening.

Descriptive (n, M [lower, upper 95% confidence intervals]) and inferential (t or Z and Cohen’s d) statistics for peak (P1, N1, P2, N2) amplitude (µV) and latency (ms) measure at sites F3/AF3 and F4/AF4 for Neuroscan versus EPOC in the active condition.

					EEG system			
Tone	ERP	Measure	Electrode	n	Neuroscan	EPOC	stat.	d	
Standard	P1	Amplitude	F3/AF3	18	3.15 [2.3, 4.0]	3.33 [2.2, 4.5]	−0.68a	0.09	
			F4/AF4	18	3.15 [2.3, 4.0]	3.09 [1.9, 4.2]	−0.6a	0.03	
		Latency	F3/AF3	18	96 [88, 104]	102 [96, 108]	−3.06a,**	0.43	
			F4/AF4	18	94 [86, 102]	104 [97, 111]	−3.58a,***	0.6	
	N1	Amplitude	F3/AF3	5	−1.72 [−3.0, −0.4]	−1.89 [−3.2, −0.6]	1.09	0.14	
			F4/AF4	5	−1.72 [−3.1, −0.4]	−2.24 [−3.8, −0.7]	2.43	0.38	
		Latency	F3/AF3	5	120 [112, 128]	136 [121, 151]	−2.52	1.48	
			F4/AF4	5	118 [110, 126]	136 [122, 150]	−3.36*	1.73	
	P2	Amplitude	F3/AF3	5	0.07 [−1.7, 1.8]	−0.08 [−1.6, 1.5]	0.62	0.09	
			F4/AF4	5	0.54 [−2.0, 3.1]	−0.09 [−1.6, 1.5]	1.14	0.3	
		Latency	F3/AF3	5	158 [133, 183]	167 [143, 191]	−3.2*	0.4	
			F4/AF4	5	167 [147, 187]	175 [159, 191]	−3.16*	0.44	
	N2	Amplitude	F3/AF3	18	−8.89 [−10.4, −7.4]	−7.93 [−9.6, −6.3]	−3.19*	0.3	
			F4/AF4	18	−8.29 [−9.6, −7.0]	−7.88 [−9.1, −6.6]	−1.34	0.15	
		Latency	F3/AF3	18	247 [238, 256]	257 [247, 267]	−5.05***	0.5	
			F4/AF4	18	248 [234, 262]	259 [246, 272]	−1.48	0.39	
Deviant	P1	Amplitude	F3/AF3	18	3.09 [1.5, 4.7]	3.07 [1.4, 4.8]	−0.3a	0.01	
			F4/AF4	18	3.07 [1.3, 4.8]	2.93 [1.1, 4.8]	−0.26a	0.04	
		Latency	F3/AF3	18	94 [85, 103]	101 [92, 110]	−3.55**	0.39	
			F4/AF4	18	93 [85, 101]	103 [95, 111]	−3.54a,***	0.58	
	N1	Amplitude	F3/AF3	4	−7.24 [−7.9, −6.6]	−6.83 [−9.0, −4.7]	−0.5	0.36	
			F4/AF4	4	−5.90 [−7.4, −4.4]	−6.89 [−8.6, −5.2]	4.08*	0.78	
		Latency	F3/AF3	4	115 [97, 133]	127 [110, 144]	−5.2*	0.82	
			F4/AF4	4	125 [102, 148]	132 [110, 154]	−15.72***	0.41	
	P2	Amplitude	F3/AF3	4	−1.74 [−3.5, −0.0]	−2.39 [−4.0, −0.8]	0.71	0.48	
			F4/AF4	4	−2.02 [−3.1, −1.0]	−2.21 [−4.7, 0.3]	0.17	0.13	
		Latency	F3/AF3	4	165 [132, 198]	186 [156, 216]	−1.85	0.81	
			F4/AF4	4	174 [143, 205]	180 [153, 207]	−1.38	0.26	
	N2	Amplitude	F3/AF3	18	−11.51 [−13.8, −9.2]	−10.00 [−12.4, −7.6]	−2.62*	0.31	
			F4/AF4	18	−10.93 [−13.0, −8.9]	−10.68 [−12.8, −8.6]	−0.43	0.06	
		Latency	F3/AF3	18	230 [213, 247]	236 [217, 255]	−1.9	0.17	
			F4/AF4	18	236 [221, 251]	238 [220, 256]	−0.26	0.05	
Notes.

a Wilcoxon Z.

* p < .05.

** p < .01.

*** p < .001.

Table 5 Neuroscan versus EPOC EEG system P3 and MMN peak comparisons.

Descriptive (n, M [lower, upper 95% confidence intervals]) and inferential (t or Wilcoxon Z and Cohen’s d) statistics for peak amplitude (µV) and latency (ms) measures produced by the Neuroscan and EPOC systems at F3/AF3 and F4/AF4 for the P3 ERP peak (to deviant tones in the active condition) and the MMN ERP component (the difference between ERPs to standard and deviant tones in the passive condition).

				EEG System			
ERP	Measure	Site	n	Neuroscan	EPOC	stat.	d	
P3	Amplitude	F3/AF3	18	−2.30 [−4.2, −0.4]	−2.46 [−4.1, −0.8]	0.45	0.04	
		F4/AF4	18	−2.51 [−4.5, −0.6]	−2.46 [−4.3, −0.6]	−0.11	0.01	
	Latency	F3/AF3	18	336 [320, 352]	351 [333, 369]	−2.91**	0.44	
		F4/AF4	18	338 [322, 354]	353 [337, 369]	−3.53**	0.45	
MMN	Amplitude	F3/AF3	18	−4.21 [−5.6, −2.9]	−4.54 [−5.8, −3.3]	0.56	0.12	
		F4/AF4	18	−4.69 [−6.1, −3.3]	−4.95 [−6.2, −3.7]	0.42	0.09	
	Latency	F3/AF3	18	193 [179, 207]	198 [181, 215]	−0.73	0.17	
		F4/AF4	18	191 [176, 206]	201 [185, 217]	−1.19	0.29	
Notes.

a Wilcoxon Z.

* p < .05.

** p < .01.

*** p < .001.

P1, N1, P2, and N2

For the P1, N1, P2, and N2 late auditory ERP peaks, there were 18 comparisons that differed statistically between the two systems. Two of the differences reflected reduced N2 amplitude in the EPOC system: differences of .99 and .93 µ Vs in the passive and active conditions respectively, both small in magnitude (d = 0.29 and .30). Sixteen of the differences reflected a delay in the latency of the peaks measured by the EPOC system and 9 of these were evident to the standard tone. The average delay for these comparisons was 8..88 ms (SD = 2.45) and effect sizes were small to large (d = 0.29 to 1.09).

P3

The differences in P3 amplitude between the systems were non-significant and small (d < 0.05). The P3 produced by the EPOC system at F3/AF3 was significantly later than that produced by the Neuroscan system by 15 ms (see Table 5). Cohen’s d effects sizes were moderate (d = 044 and 0.45).

MMN

The differences in MMN amplitude and latency between the two systems were non-significant and small (d < 0.03).

Discussion

The aim of the current study was to assess the validity of the Emotiv EPOC gaming EEG system as an auditory ERP measurement tool in children. To this end, we simultaneously measured ERPs using a research-grade Neuroscan system and the EPOC system in children aged between 6 and 12 years. Children were presented with standard and deviant tones in both passive (ignore tones) and active (count high tones) listening conditions. There are three key findings. First, whilst both EEG systems recorded a high proportion of accepted epochs, fewer were acceptable for EPOC. This was also found by Badcock et al. (2013) when they tested adults. Fewer acceptable epochs with EPOC may stem from reduced stability of EPOC’s saline-soaked cotton sensors resting on the scalp, relative to the gel used with Neuroscan, which effectively glues to sensor the scalp with gel. Having said this, EPOC recorded adequate numbers of acceptable epochs to produce reliability later auditory ERPs.

Second, the systems produced similar late auditory ERP (ICCs: 0.82–0.95) and MMN waveforms (ICCs: 0.67–0.74). These ICCs are higher than those previously reported by Badcock et al. (2013) who found that ICCs in adults for the late auditory ERPs ranged from 0.57 to 0.80, and that the ICC for the MMN was 0.44. Apart from differences in populations studied (i.e., children versus adults), there are three major differences between the current and the previous study that may explain the different ICCs. First, the current study was conducted within a shielded room, whereas the previous study was not. It is conceivable that the shielded room resulted in cleaner EEG recordings. Second, the analysis for the current study used only overlapping epochs that were accepted by both EEG systems. However, the previous study rejected fewer epochs, which would have resulted in a high proportion of overlapping epochs. Thus, this seems an unlikely explanation of the higher ICCs in the current study. Third, the current study used a wireless triggering system for the EPOC system, while the previous study used a wired system. In the previous study, we noted that participant movement with the wired system caused interference to the EEG signal. However, fewer epochs were rejected in the previous study so this too seems an unlikely explanation for the higher ICCs in the current study. It therefore seems most likely that the shielded room produced the higher ICCs in the current study compared to the previous study done with adults by Badcock et al.

Third, there were only a few differences between the peak amplitude and latency measures produced by the EPOC and Neuroscan systems, which mostly related to delayed latencies for the EPOC system (i.e., an average delay was 8.1 ms (SD = 5.92)). This represents a single sample at 128 Hz. Since this delay was small, and occurred in a minority of comparisons, we do not believe it significantly compromises the use of the EPOC system as a measure of auditory P1, N1, P2, N2, or P3 ERPs in children.

Overall, the findings of the present study paired with Badcock et al. (2013) suggest that EPOC compares well with Neuroscan for investigating late auditory ERPs in children. This opens up new opportunities for conducting ERP studies with children with or without cognitive impairments who find the laboratory settings associated with traditional research-grade EEG systems threatening or uncomfortable. It also paves the way for large-scale studies of the development of typical and atypical ERPs since it allows the measurement of children’s ERPs in settings such as schools, childcare centres, hospitals, and private clinical practices.

Supplemental Information

Figure S1 Individual auditory ERP waveforms in the passive listening condition

The auditory ERP waveforms of individuals (grey lines) at F3/AF3 and F4/AF4 for standard (panels A, B, E & F) and deviant stimuli (panels C, D, G, & H) in the passive condition for the Neuroscan (left; panels A, C, E, & G) and EPOC (right; panels B, D, F & H) systems. The black line represents mean and the bold grey are the 95% confidence intervals.

Click here for additional data file.

Figure S2 Individual auditory ERP waveforms in the active listening condition

The auditory ERP waveforms of individuals (grey lines) at F3/AF3 and F4/AF4 for standard (panels A, B, E & F) and deviant stimuli (panels C, D, G, & H) in the passive condition for the Neuroscan (left; panels A, C, E, & G) and EPOC (right; panels B, D, F & H) systems. The black line represents mean and the bold grey are the 95% confidence intervals.

Click here for additional data file.

Figure S3 Individual auditory Mismatch Negativity (MMN) waveforms

The MMN waveforms of individuals (grey lines) at F3/AF3 (panels A & B) and F4/AF4 (panels C & D) for standard and deviant stimuli in the passive condition for the Neuroscan (left; panels A & C) and EPOC (right; panels B & D) systems. The black line represents mean and the bold grey are the 95% confidence intervals.

Click here for additional data file.

We would like to thank the participants and their parents who volunteered their time.

Additional Information and Declarations

Competing Interests

Author Contributions

Human Ethics

Genevieve McArthur is an Academic Editor for PeerJ. Katharine Glenn works for MultiLit, a literacy instruction enterprise which is not a competing interest to the current research. None of the other authors have competing interests.

Nicholas A. Badcock conceived and designed the experiments, performed the experiments, analyzed the data, contributed reagents/materials/analysis tools, wrote the paper, prepared figures and/or tables, reviewed drafts of the paper.

Kathryn A. Preece performed the experiments, analyzed the data, wrote the paper, prepared figures and/or tables, reviewed drafts of the paper, managed recruitment, scheduling, and personnel.

Bianca de Wit and Katharine Glenn performed the experiments, analyzed the data, wrote the paper, reviewed drafts of the paper.

Nora Fieder performed the experiments, analyzed the data, reviewed drafts of the paper.

Johnson Thie contributed reagents/materials/analysis tools, wrote the paper, reviewed drafts of the paper.

Genevieve McArthur conceived and designed the experiments, wrote the paper, reviewed drafts of the paper.

The following information was supplied relating to ethical approvals (i.e., approving body and any reference numbers):

The Macquarie University Human Research Ethics Committee approved the methods used in this study (approval number: 5201200658).

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
