# Peer review of "Validation of the Emotiv EPOC EEG system for research quality auditory event-related potentials in children"

_PeerJ, doi:10.7717/peerj.907_

## Round 0.1 · original submission · Major Revisions

· Academic Editor

Major Revisions

Dear Authors,There are numerous major issues to settle before this manuscript is ready to be re-reviewed.Please read carefully the comments and make the changes that are considered extremely important.

·

Basic reporting

The current manuscript by Badcock and colleagues describes a study on the feasibility of recording auditory ERPs with a mobile EEG system (EPOC) in children. They present a passive and an active auditory oddball block to subjects wearing both a research and a mobile EEG system. Their results show clearly that obligatory responses (here specifically long-latency responses such as P1, N1, P2, N2) are reliably recordable with the mobile system. However, the mismatch negativity failed to show the same reliability (even though amplitude and latency were quite comparable).

The manuscript is well written, the methods and analyses are state of the art. Furthermore, the results are important and will be the basis for future research in developmental cognitive neuroscience. I have only minor comments that should be addressed before publication (see below).

Experimental design

The following comments could also be under 'Validity of the Findings'

Methods: Please specify filter settings (e.g., type, cut-off).

Methods, Offline EEG processing: “This process failed to identify any eye-blink related components for any individual dataset.” Can this sentence be simplified? It took me two to three times reading it to understand it (actually the next sentence only made it clear).
Also, related to this: why is the ICA reported if it does not aid to the analysis?

Methods/Results: The authors report that less trials remained for the EPOC system. How large was the overlap between the two datasets? Could this influence the results? I think the authors may think about reducing their comparison to the overlapping trial sample. This would make the claim even stronger (after all this is a feasibility check on ‘exactly’ the same data).

Methods: I think the intraclass correlations need to be explained more in detail, or references to papers using ICCs on electrophysiology data need to be given. On that note, I am a little confused that indeed the MMN is so much 'different' from the obligatory responses. And simply stating that the MMN is an unstable measure is not enough, as the data should be highly correlated given that they are measured from the same person at the same time. The MMN is quite variable across subjects and even recording sessions, that much is true, but this weak of a correlation at the same time in the same person? Also considering that standard and deviant are so highly correlated? Do the authors have any further points that could add to the discussion of this finding?

Validity of the findings

No further comments

Additional comments

Introduction: “Unfortunately, no study has yet tested if an adapted EPOC system can produce valid auditory ERPs in children.” I wouldn’t say ‘unfortunately’. The field of mobile EEG research is still new, so there are still a lot of open questions. Also for the authors it is not so unfortunate.

Discretionary remarks:

In the abstract (Results) the P2 is named twice.

Artwork: The authors could consider presenting topographies from the two systems, that would add some color and be informative as to how well comparable they are.

One of the affiliations is not taken by any of the authors (#4).

Reviewer 2 ·

Basic reporting

The authors compare two different EEG headsets for the auditory ERP in children. One headset is the high end research focused headset (Neuroscan) while the other is the cheap off the shelf gaming headset (Emotiv). The authors try to show that the Emotiv headset produce similar results as Neuroscan.

Experimental design

The following are the main comments on the experiment design:
(I) Authors did not discuss the criteria which is important for such a comparison. They first need to spell out in detail the criteria for the comparison, for example, the importance of sampling rate
(2) As a result, the authors fail to understand the importance of various parameters. For example, higher sampling rate allows more temporal information to be collected. Hence, it is an advantage to have higher sampling rate. For Emotiv, its 128 hz while Neuroscan can go higher. Sampling rate of 128 hz means that every sample is collected about after 8msec. Hence, there is significant data loss, especially for auditory components where auditory ERPS are observed in the range of 20 to 40ms. Hence its a major limitation in data acquisition from Emotiv. The Neuroscan can give better sampling rate. Unfortunately, authors downsampled it to 128 hz to match, which is completely wrong. Of course the results will be similar. However, that does not mean that they will be correct. Neuroscan has the advantage of higher sampling rate which authors ignore completely.
(3) In addition, they do not discuss other important parameters like sample resolution. It appears that authors are not aware of the significance of these parameters
(4) Authors failed to identify the various artifacts including eye blinks, eye movement, muscular, ECG. That is a bad experiment design.

Validity of the findings

The experiment design has major flaws. So the results cannot be verified.

Additional comments

(1) What is adapted experiment design? Its not clear at all. Authors should use figures to explain and clearly show their contribution.
(2) Placement of electrodes on the scalp should be clearly shown in figures.

Reviewer 3 ·

Basic reporting

• This article presents a comparative study to validate the performances of EPOC EEG system using Auditory ERP as a domain. The flow of this article is coherent to the title. The authors did provide basic introduction to their work but i recommend to add more important aspects in justifying the rationale of this work.

Experimental design

• As for the experimental parts, 19 subjects are not good enough to represent probable validity of the experiments. I would like to know, is there any statistical basis to explain the number of selected subjects, or did I miss something fundamental here?
• This is equality important to provide some statistical significant to choose any sample size values (i.e usability test sample might be different as opposed to the survey sample size)
• The choice of baseline device and software. The authors must address some basis to justify his/her ideas. I would recommend conducting more experiments using different devices to validate the results.

Validity of the findings

• Based on this statement “ Regarding to (3), peak amplitude and latency measures were initially calculated using an automated procedure that identified the point of maximum amplitude (positive or negative) within appropriate time intervals, determined by visual inspection of the relevant grand mean” (line 159-161), - Could you provide any literature to support this method? I find it quite arbitrary.
• The experimental results have confirmed a number of differences between these two devices for specific cases. However the authors did not explain technical differences that may cause these discrepancies.

---

## Round 0.2 · accepted · Accept

· Academic Editor

Accept

Dear Authors,The manuscript that has been revised is now accepted and will undergo PeerJ processing for publication.Thank you for your submission to PeerJ.

·

Basic reporting

--

Experimental design

--

Validity of the findings

--

Additional comments

I think the authors have done an excellent job in revising the manuscript. In my view, this is an important study and I would like to congratulate the authors on this work.